# Influence of the Digital Mock-Up and Experience on the Ability to Determine the Prosthetically Correct Dental Implant Position during Digital Planning: An In Vitro Study

**DOI:** 10.3390/jcm9010048

**Published:** 2019-12-24

**Authors:** Miriam O’Connor Esteban, Elena Riad Deglow, Álvaro Zubizarreta-Macho, Sofía Hernández Montero

**Affiliations:** Department of Implant Surgery, Faculty of Health Sciences, Alfonso X el Sabio University, 28691 Madrid, Spain; miriam_oconnor5@hotmail.com (M.O.E.); elenariaddeglow@gmail.com (E.R.D.); shernmon@uax.es (S.H.M.)

**Keywords:** digital wax, digital mock-up, guided implant surgery, digital implant planning, implantology

## Abstract

The purpose of this study was to analyze the influence of the digital mock-up and operator experience on the dental implant planning position. A total of 200 dental implants were planned, which were distributed into two study groups: A. dental implant planning by dental surgeons with 5–10 years of experience (*n* = 80); and B. dental implant planning by dental surgery students without experience (*n* = 120). Operators were required to plan eight dental implants in the same maxillary edentulous case uploaded in 3D implant-planning software, before and after using the digital mock-up. Deviations between the dental implant planning positions before and after using the digital mock-up were analyzed at platform, apical and angular levels, and were analyzed using a 3D implant-planning software using Student’s *t* test. The paired *t*-test revealed statistically significant differences between the deviation levels of participants with 5–10 years’ experience and no experience at the platform, apical and angular levels. Digital mock-ups allow for more accurate dental implant planning regardless of the experience of the operator. Nevertheless, they are more useful for operators without dental surgery experience.

## 1. Introduction

Three-dimensional (3D) implant positioning is a challenge when providing edentulous patients with dental implants [1]. Deviations and inaccuracy in dental implants’ positioning can cause complications in the subsequent prosthetic restoration that are difficult to solve [2]. In addition, the transfer of the planned implant position information to the clinical scene has evidenced inaccuracy, which can cause risk and complications [3]. Implant placement using image-data-based navigation has been introduced to the field of dental implants in an attempt to improve the prosthetically correct positioning of dental implant placement and avoid the potential risks of this therapeutic procedure [4]. These surgical approaches are based on modern 3D implant treatment planning systems, based on digital datasets obtained from a Cone Beam Computed Tomography (CBCT) scan and a digitized dental surface, and improve the accuracy of dental implant placement by means of a virtual planning simulation [5]. Nevertheless, the dental implant planning position is affected by the level of professional experience [6]. In addition, the 3D implant treatment planning systems are mainly based on anatomical considerations, regardless of prosthetic needs [7]. The digital mock-up offers valuable information related to the occlusal plane and aesthetics, allowing prosthetically guided surgeries, and defining the prosthetic alternatives at an early stage [8]. Moreover, it offers useful information to evaluate the necessity of bone augmentation techniques [9]. A correct combination of anatomical information and virtual prosthetic planning allows the positioning of the implant according to the prosthetic’s needs [10]. The incorporation of the digital mock-up into the dental implant planning process and digital flow in implantology can influence the accuracy of dental implant placements and reduce the variability associated with the professional experience [11].

The aim of this work was to analyze and compare the accuracy of dental implants’ planning depending on the experience of the operator, with the aid of a digital mock-up, with a null hypothesis (H0) stating that there would be no difference between the experience of the operator with the aid of the digital mock-up with regards to the accuracy of dental implant planning.

## 2. Materials and Methods

### 2.1. Study Design

This in vitro experiment was performed at the Dental Centre of Innovation and Advanced Specialties at Alfonso X El Sabio University (Madrid, Spain) between March and July 2019. The patient gave his consent to provide the DICOM files from the CBCT scan (WhiteFox, Acteón Médico-Dental Ibérica S.A.U.-Satelec, Merignac, France) and the STL file from the extraoral scan (EVO, Ceratomic, Protechno, Girona, Spain).

### 2.2. Experimental Procedure

Two hundred dental implants (BioHorizons; Birmingham, AL, USA) were planned in tooth positions 2.6, 2.5, 2.3, 2.2, 1.2, 1.3, 1.5, and 1.6 (4.6 × 12 mm, conical wall and internal taper) in a virtual model of a totally edentulous upper jaw (Sawbones Europe AB; Malmo, Sweden), obtained from a real clinical case. The case selected presented a high amount of available bone and was scheduled for a fixed implant-supported prosthesis. The datasets were uploaded and aligned in a 3D implant-planning software (NemoStudio^®^, Nemotec; Madrid, Spain) to allow dental implant planning by the following study groups: A. dental implant planning by dental surgeons with 5–10 years of experience (*n* = 80); and B. dental implant planning by dental surgery students without experience (*n* = 120). The dental implant planning was performed in a 3D implant-planning software (NemoStudio^®^, Nemotec; Madrid, Spain) by each operator before (Figure 1A–D) and after (Figure 1E–H) uploading and aligning the digital mock-up, which was obtained after scanning the complete denture of the patient, to evaluate the influence of the digital mock-up on the dental implant planning position.

The following deviations between the dental implant planning positions before and after using the digital mock-up were analyzed: platform deviation, measured at the entry point; apical deviation, measured at the apical endpoint; and angular deviation, measured in the center of the cylinder. Deviations of all dental implant planning were evaluated and compared in the axial (Figure 2B,E), sagittal (Figure 2A,D) and coronal (Figure 2C,F) views by the same expert operator, and the results were expressed in each position.

### 2.3. Statistical Analysis

A statistical analysis of all variables was carried out using SAS 9.4 (SAS Institute Inc., Cary, NC, USA). Descriptive statistics were expressed as means and standard deviation (SD) for quantitative variables. A comparative analysis was performed by comparing the mean deviation values between the implant positions planned with and without the mock-up using the Student’s *t*-test, as the variables had a normal distribution. The statistical significance was set at *p* ˂ 0.05.

## 3. Results

Table 1 shows the means and SD values for deviation. A mean deviation of 1.20 ± 0.98 mm (min: 0.00 mm, max: 3.50 mm) and 1.83 ± 1.60 mm (min: 0.00 mm, max: 8.60 mm) was observed at the platform of the 5–10 years’ experience (Figure 2A–C) and no experience (Figure 2D–F) study groups, respectively (Figure 3).

The paired *t*-test revealed statistically significant differences between the platform deviations of 5–10 years’ experience and the no experience study groups *(p ˂* 0.001). A mean deviation of 0.93 ± 1.02 mm (min: 0.00 mm, max: 3.60 mm) and 1.90 ± 1.90 mm (min: 0.00 mm, max: 11.90 mm) was observed at the apical endpoint of the 5–10 years’ experience (Figure 2A–C) and no experience (Figure 2D–F) study groups, respectively (Figure 4).

The paired *t*-test revealed statistically significant differences between the apical deviations of participants with 5–10 years’ experience and no experience *(p ˂* 0.001). A mean angular deviation of 2.06° ± 2.17° (min: 0.10°, max: 8.20°) and 3.73° ± 3.88° mm (min: 0.20°, max: 21.70°) was observed in the 5–10 years’ experience (Figure 2A–C) and no experience (Figure 2D–F) study groups, respectively (Figure 5). The paired *t*-test revealed statistically significant differences between the angular deviations of 5–10 years’ experience and no experience study groups *(p =* 0.002).

## 4. Discussion

The results obtained in the present study rejected the null hypothesis (H0) that states that there would be no difference between the experience of the professional after using the digital mock-up with regard to the accuracy of dental implant planning. The mock-up technique demonstrated its efficacy in transferring the information from the diagnostic wax-up to the patient’s mouth. A more accurate diagnosis and more thorough treatment planning lead to a safer, more predictable and conservative treatment. Therefore, it has been extensively used in restorative and aesthetic treatments to improve the result for the patient. In the field of dental surgery, it has been used to perform guided dental implant surgery and crown lengthening [12]. It has also used as a communication tool between the dentist, patient and technician [13]. Digital mock-ups allow a useful virtual pre-operative planning of the functional and aesthetic aspects of the procedure, avoiding possible intra-operative complications [14]. A wide range of factors can affect the dental implant placement position, such as prosthodontically-driven treatment planning, site preparation, the surgeon’s experience and the use of a surgical guide. The surgeon’s experience has been demonstrated to be a relevant prognosis factor, to the point of reducing patients’ subsequent complications [15]. Recently, dental implant placement using image-data-based navigation has been introduced to the field of dental surgery in an attempt to improve the accuracy of dental implant placement and avoid the potential risks associated with this therapeutic procedure. The use of surgical computer-aided static navigation systems significantly improved (*p* ˂ 0.0001) the accuracy of dental implant placement, regardless of the surgeon’s experience [16]. Furthermore, the use of surgical computer-aided dynamic navigation systems also significantly improved (*p* ˂ 0.0001) the accuracy of the dental implant placement position in surgeons without experience [17]. Both techniques are sustained on a preoperative planning procedure, based on a CBCT scan and an intra/extraoral scan, which can affect the accuracy of the dental implant placement position, but these techniques require a large learning curve [18]. Another factor that affects the position of dental implant placement is the presence/absence of tooth structures that can guide the position and angulation of the dental implant. A totally edentulous jaw represents a challenge to the clinician because it makes the dental implant planning position difficult. The use of a digital mock-up allows accurate, prosthesis-based planning; the fabrication of a surgical template and also a provisional prosthesis. The results show that all the professionals considered changing the dental implant planning position after using the digital mock-up, regardless of the study group. Nevertheless, the “no experience” study group needed to perform more changes on the dental implant planning after using the digital mock-up. This implies that the digital mock-up helped to find a more prosthetically correct position of dental implant planning for the subsequent prosthetic treatment, regardless of the experience of the professional, but it is more useful for professionals with less surgical experience. The results obtained in this study also demonstrate the influence of the digital mock-up on the implant planning position, regardless of the surgeon’s experience. The 5–10 years’ experience study group considered it necessary to change the dental implant planning position by a mean of 1.02 mm to the platform level, 0.93 mm apically and 2.06° angulation, and the no experience study group changed the dental implant planning position by a mean of 1.83 m to the platform level, 1.90 mm apically and 3.73° angulation.

The teaching objective derived from this study is that the digital mock-up allows for more accurate dental implant planning. Nevertheless, further research is needed to determine the influence of dental implant planning with a digital mock-up on the accuracy of dental implant placements and potential clinical complications.

## 5. Conclusions

In conclusion, within the limitations of this in vitro study, the results show that planned implant positions using the digital mock-up are closer to the prosthetically ideal implant position, regardless of the experience of the operator. Nevertheless, it is more useful to operators without dental surgery experience.

## Figures and Tables

**Figure 1 jcm-09-00048-f001:**
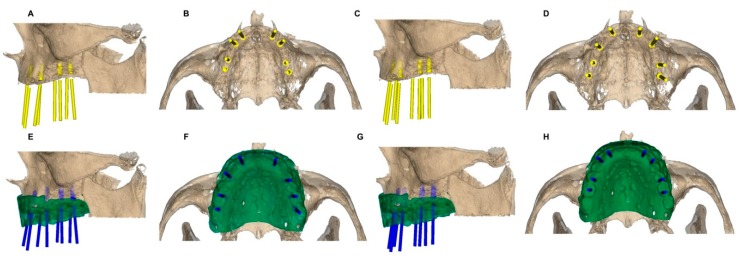
Dental implant planning before (**A**–**D**) and after (**E**–**H**) using the digital mock-up in 5–10 years’ experience (**A**,**B**,**E**,**F**) and no experience (**C**,**D**,**G**,**H**) study groups.

**Figure 2 jcm-09-00048-f002:**
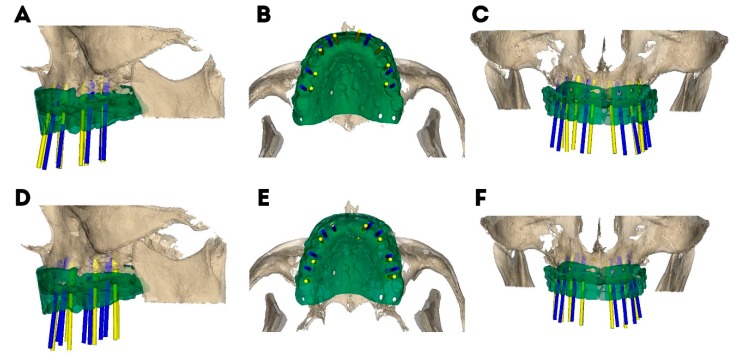
Dental implant planning before (yellow cylinders) and after (blue cylinders) using the digital mock-up in both study groups (5–10 years’ experience (**A**–**C**) and no experience (**D**–**F**)).

**Figure 3 jcm-09-00048-f003:**
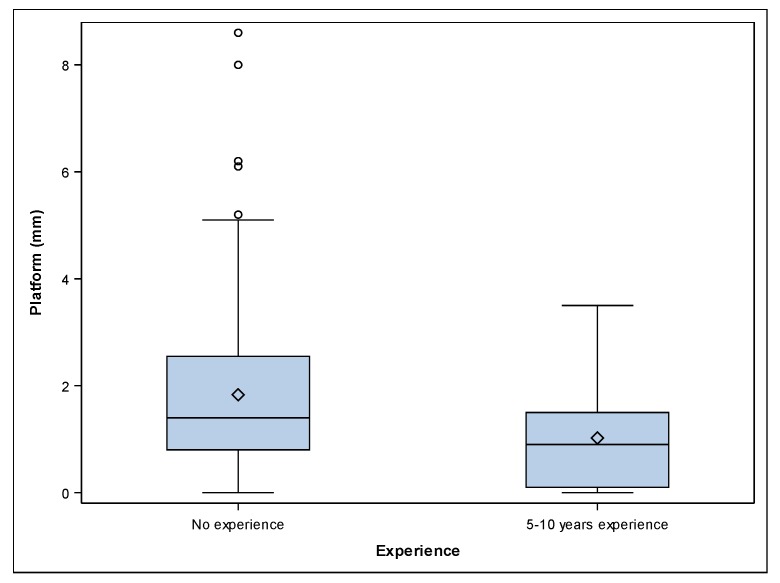
Box plots of platform deviations of the experimental groups. The horizontal line in each box represents the median value.

**Figure 4 jcm-09-00048-f004:**
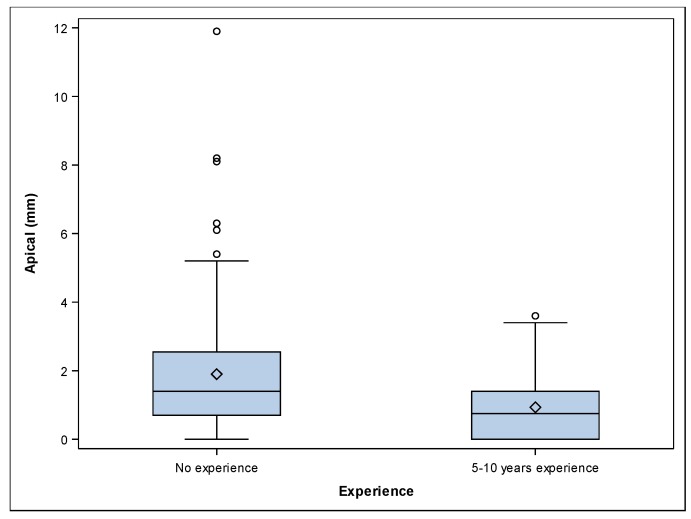
Box plots of apical deviations of the experimental groups. The horizontal line in each box represents the median value.

**Figure 5 jcm-09-00048-f005:**
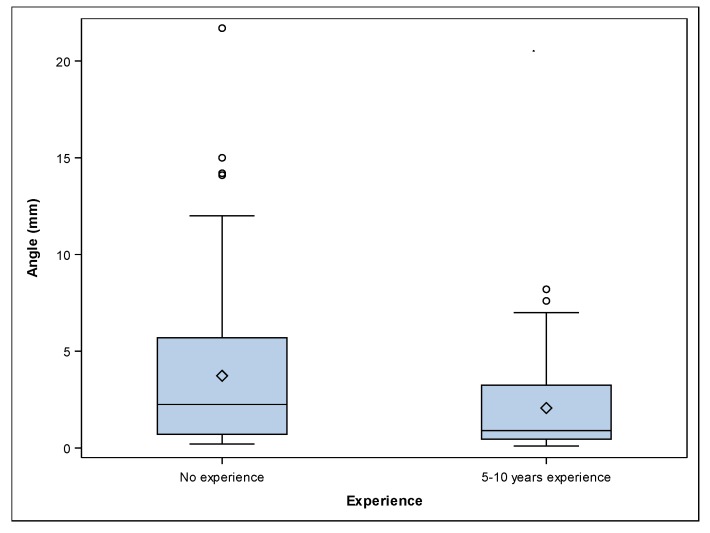
Box plots of angular deviations of the experimental groups. The horizontal line in each box represents the median value.

**Table 1 jcm-09-00048-t001:** Descriptive deviation values at the platform (mm), apical (mm) and angular levels in both study groups (5–10 years’ experience and no experience) after using the digital mock-up.

		*n*	Mean	SD	Minimum	Maximum
PLATFORM	5–10 years’ experience	80	1.02	0.98	0.00	3.50
	No experience	120	1.83	1.60	0.00	8.60 *
APICAL	5–10 years’ experience	80	0.93	1.02	0.00	3.60
	No experience	120	1.90	1.90	0.00	11.90 *
ANGULAR	5–10 years’ experience	80	2.06	2.17	0.10	8.20
	No experience	120	3.73	3.88	0.20	21.70 *

*: statistically significant differences.

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
