# Peer review of "Influence of the Digital Mock-Up and Experience on the Ability to Determine the Prosthetically Correct Dental Implant Position during Digital Planning: An In Vitro Study"

_jcm, 2019, doi:10.3390/jcm9010048_

Round 1

Reviewer 1 Report

I would like to congratulate the authors on their work. Prosthetically driven implant placement is paramount to achieve long term success in implant dentistry. Incorrect positioning of dental implants may compromise the white and pink aesthetics of the implant prosthesis and may lead to biomechanical and biological complications. The benefits of navigated implant placement include shorter surgical time, the prevention of surgical complications, and in case of a flapless approach lower postoperative morbidity. However, the most significant advantage of navigated implant placement is the opportunity to carry out a prosthetically driven approach by aligning the three-dimensional data of the mock-up to the CBCT data. The manuscript investigates whether the experience of the surgeon and knowledge of the three-dimensional position of the future prosthesis might facilitate correct planning of the implants.

The exact ideal implant position for the prosthesis is hard to reproduce. For any given prosthesis there is a range of acceptable implant positions. The methods of this study assume that participants in both groups found a position within the range of acceptable positions after seeing the digital mock-up. In my opinion this assumption and its validation should be clarified in the discussion section of the manuscript. According to the methods of the study operators were required to plan the implants in the 2.6, 2.5, 2.3, 2.2, 1.2, 1.3, 1.5 and 1.6 tooth positions. This should be supported by the figures. However, in the specific instance given in Figure 2. The inexperienced participant did not manage to place the implants in these tooth positions. I suggest finding a better illustration for Figure 2.

In the manuscript the authors refer to the participant’s ability to correct his or her implant position after receiving the mock-up as accuracy. However, in the literature of digital dentistry accuracy is often referred to as the difference of planned and executed implant positions. To avoid misreading I suggest rephrasing these parts of the manuscript so that accuracy is swapped to the ability to determine prosthetically correct implant positions.

Title

I suggest changing the title to “Influence of the digital mock-up and experience on the ability to determine the prosthetically correct dental implant position during digital planning: An in vitro study.”

Abstract

In the abstract I suggest rephrasing row 10. To “The purpose of the study.” and including the operators experience.

I recommend rephrasing the abstract to include that the operators were required to plan 8 dental implants in the same maxillary edentulous case. P levels may be set at either 0,05 or 0,005. Accurate display of p levels in the abstract is not necessary.

Instructions for authors in the Journal dictate, that headings like purpose, materials, etc. should be left out.

Introduction

I suggest revising the statement: “In addition, the 3D implant treatment planning systems are only based on anatomical considerations, regardless of prosthetic needs.” in row 37 because most implant planning software allows for prosthetic driven treatment planning.

In row 45 I recommend rephrasing accuracy to prosthetically correct position.

Materials and Methods

Study design

Please explain why ISO 14801 standard is required for this study

Experimental procedure

I recommend moving the patient consent part to the study design section.

I propose detailing the kind of prostheses planned and the amount of bone available (stage of atrophy), because methods of retention (cemented or screw-retained) and the need for a hybrid prosthesis instead of a bridge might affect the planned position of the implants and the need for bone augmentation.

I recommend finding a better illustration for Figure 2.

Statistical analysis

In row 86: “Descriptive statistics were expressed as means and standard deviation (SD) for quantitative variables and as absolute numbers and percentages for qualitative variables” What kind of qualitative variables were used in the statistical analysis? If there weren’t any this part should be rephrased.

In row 88 I suggest rephrasing planned and performed implant placement to implant positions planned without the mock-up and with the mock-up.

Row 117-122 belongs to the discussion section. I recommend revising this statement according to row 45.

Conclusions

I suggest rephrasing the conclusion so that instead of using accuracy it should state that planned implant positions using the digital mock-up are closer to the prosthetically ideal implant position.

References

Instructions for authors for the Journal dictate that journal articles should be referred to in the following format: Author 1, A.B.; Author 2, C.D. Title of the article. Abbreviated Journal Name YearVolume, page range. However, in the manuscript in some references names of the authors are not abbreviated or abbreviated with a different punctuation. Please revise accordingly.

Summary

The manuscript is dealing with the relevant issue of digital treatment planning. However, some methodical issues should be clarified before publication. In my opinion the manuscript should be submitted to extensive language editing.

Author Response

Dear reviewer,

We are pleased to resubmit the manuscript of the work entitled, “Influence of the digital mock-up and experience on the ability to determine the prosthetically correct dental implant position during digital planning: An in vitro study”.

Reviewer 1: Extensive editing of English language and style required.

Response: In order to adapt to the reviewer's 1 comments, we have send the manuscript to a specialized traductor.

Reviewer 1: Title: I suggest changing the title to “Influence of the digital mock-up and experience on the ability to determine the prosthetically correct dental implant position during digital planning: An in vitro study.”

Response: In order to adapt to the reviewer's 1 comments, we have changed the title.

Reviewer 1: Abstract: I suggest rephrasing row 10. To “The purpose of the study.” and including the operators experience.

I recommend rephrasing the abstract to include that the operators were required to plan 8 dental implants in the same maxillary edentulous case. P levels may be set at either 0,05 or 0,005. Accurate display of p levels in the abstract is not necessary.

Instructions for authors in the Journal dictate, that headings like purpose, materials, etc. should be left out.

Response: In order to adapt to the reviewer's 1 comments, we have rephrasing the Abstract.

Reviewer 1: Introduction: I suggest revising the statement: “In addition, the 3D implant treatment planning systems are only based on anatomical considerations, regardless of prosthetic needs.” in row 37 because most implant planning software allows for prosthetic driven treatment planning.

In row 45 I recommend rephrasing accuracy to prosthetically correct position.

Response: In order to adapt to the reviewer's 1 comments, we have we have rephrasing the paragraph.

Reviewer 1: Materials and Methods. Study design: Please explain why ISO 14801 standard is required for this study.

Response: The ISO 14801 standard is no required, so we have removed the paragraph.

Reviewer 1: Materials and Methods. Experimental procedure: I recommend moving the patient consent part to the study design section.

Response: In order to adapt to the reviewer's 1 comments, we have moved this paragraph from “Experimental procedure” section to “Study design” section.

Reviewer 1: Materials and Methods. Experimental procedure: I propose detailing the kind of prostheses planned and the amount of bone available (stage of atrophy), because methods of retention (cemented or screw-retained) and the need for a hybrid prosthesis instead of a bridge might affect the planned position of the implants and the need for bone augmentation.

Response: In order to adapt to the reviewer's 1 comments, we have added a paragraph at “Experimental procedure” section.

Reviewer 1: I recommend finding a better illustration for Figure 2.

Response: In order to adapt to the reviewer's 1 comments, we have changed the Figure 2 and adapted the references to Figure 2.

Reviewer 1: Statistical analysis: In row 86: “Descriptive statistics were expressed as means and standard deviation (SD) for quantitative variables and as absolute numbers and percentages for qualitative variables” What kind of qualitative variables were used in the statistical analysis? If there weren’t any this part should be rephrased.

Response: In order to adapt to the reviewer's 1 comments, we have rephrased this paragraph.

Reviewer 1: Statistical analysis: In row 88 I suggest rephrasing planned and performed implant placement to implant positions planned without the mock-up and with the mock-up.

Response: In order to adapt to the reviewer's 1 comments, we have rephrase this paragraph.

Reviewer 1: Statistical analysis: Row 117-122 belongs to the discussion section. I recommend revising this statement according to row 45.

Response: In order to adapt to the reviewer's 1 comments, we have rephrased this paragraph and added to the Discussion section.

Reviewer 1: Conclusions: I suggest rephrasing the conclusion so that instead of using accuracy it should state that planned implant positions using the digital mock-up are closer to the prosthetically ideal implant position.

Response: In order to adapt to the reviewer's 1 comments, we have rephrase this paragraph.

Reviewer 1: References: Instructions for authors for the Journal dictate that journal articles should be referred to in the following format: Author 1, A.B.; Author 2, C.D. Title of the article. Abbreviated Journal Name YearVolume, page range. However, in the manuscript in some references names of the authors are not abbreviated or abbreviated with a different punctuation. Please revise accordingly.

Response: In order to adapt to the reviewer's 1 comments, we have rephrase the references.

We take this opportunity to thank the recommendations and suggestions made by the reviewer to improve the document.

Yours sincerely,

Álvaro Zubizarreta Macho, DDS, PhD.

Avda. de la Universidad, 1. 28691. Villanueva de la Cañada. Madrid. Spain.

E-mail: amacho@uax.es

Reviewer 2 Report

line 35: please define CBCT.

line 75-77: please check punctuation. Punctuation makes the sentence difficult to understand.

line 78 and Figure 2. in the text, a coronal view is mentioned, but is missing in figure 2. Please tune the text and the figure.

line 95: please check the references to figure 2. no experience group is Figure 2C-D.

line 106, 114: as in line 95

line 132: missing punctuation

line 158: please revise "1.m83m"

line 160: please revise "mock up"

Author Response

Dear reviewer,

We are pleased to resubmit the manuscript of the work entitled, “Influence of the digital mock-up and experience on the ability to determine the prosthetically correct dental implant position during digital planning: An in vitro study”.

Reviewer 2: I don't feel qualified to judge about the English language and style.

Response: In order to adapt to the reviewer's 2 comments, we have send the manuscript to a specialized traductor.

Reviewer 2: line 35: please define CBCT.

Response: In order to adapt to the reviewer's 2 comments, we have defined the abbreviation.

Reviewer 2: line 75-77: please check punctuation. Punctuation makes the sentence difficult to understand.

Response: In order to adapt to the reviewer's 2 comments, we have sent the manuscript to a specialized translator who has reviewed the style and punctuation.

Reviewer 2: line 78 and Figure 2. in the text, a coronal view is mentioned, but is missing in figure 2. Please tune the text and the figure.

Response: In order to adapt to the reviewer's 2 comments, we have changed Figure 2 and rephrase this paragraph.

Reviewer 2: line 95: please check the references to figure 2. no experience group is Figure 2C-D.

Response: In order to adapt to the reviewer's 2 comments, we have rephrase this paragraph.

Reviewer 2: line 106, 114: as in line 95.

Response: In order to adapt to the reviewer's 2 comments, we have rephrase this paragraph.

Reviewer 2: line 132: missing punctuation.

Response: In order to adapt to the reviewer's 2 comments, we have sent the manuscript to a specialized translator who has reviewed the style and punctuation.

Reviewer 2: line 158: please revise "1.m83m".

Response: In order to adapt to the reviewer's 2 comments, we have changed the word.

Reviewer 2: line 160: please revise "mock up".

Response: In order to adapt to the reviewer's 2 comments, we have changed the word.

We take this opportunity to thank the recommendations and suggestions made by the reviewer to improve the document.

Yours sincerely,

Álvaro Zubizarreta Macho, DDS, PhD.

Avda. de la Universidad, 1. 28691. Villanueva de la Cañada. Madrid. Spain.

E-mail: amacho@uax.es